# Survival and Prognostic Factors of Ultra-Central Tumors Treated with Stereotactic Body Radiotherapy

**DOI:** 10.3390/cancers14235908

**Published:** 2022-11-29

**Authors:** Viola Salvestrini, Marloes Duijm, Mauro Loi, Joost J. Nuyttens

**Affiliations:** 1CyberKnife Center, Istituto Fiorentino di Cura ed Assistenza, 50139 Florence, Italy; 2Radiation Oncology Unit, Azienda Ospedaliero-Universitaria Careggi, 50134 Florence, Italy; 3Department of Radiation and Oncology, Erasmus MC-Cancer Institute, 3015 GD Rotterdam, The Netherlands

**Keywords:** ultra-central lung tumors, SBRT, acute and late toxicity

## Abstract

**Simple Summary:**

Stereotactic body radiotherapy for tumors directly touching or overlapping the trachea, proximal bronchial tree, or esophagus may be a high-risk clinical scenario. Some prognostic factors may be associated with better survival for ultra-central (UC) tumors. Indeed, the present study reports that patients with larger tumor sizes, who are elderly, and male, treated for ultra-central lung tumors, report a worse significant overall survival [OS: 33 vs. 15 months (*p* 0.028), 40 vs. 23 months (*p* 0.005), 40 vs. 24 months (*p* 0.027), respectively]. Tumor locations further away from the hilus of the lung is associated with a better prognosis.

**Abstract:**

Introduction: Stereotactic body radiotherapy (SBRT) reported excellent outcomes and a good tolerability profile in case of central lung tumors, as long as risk-adapted schedules were adopted. High grade toxicity was more frequently observed for tumors directly touching or overlapping the trachea, proximal bronchial tree (PBT), and esophagus. We aim to identify prognostic factors associated with survival for Ultra-Central (UC) tumors. Methods: We retrospectively evaluated patients treated with SBRT for primary or metastatic UC lung tumors. SBRT schedules ranged from 45 to 60 Gy. Results: A total number of 126 ultra-central lung tumors were reviewed. The Median follow-up time was 23 months. Median Overall Survival (OS) and Progression Free Survival (PFS) was 29.3 months and 16 months, respectively. Local Control (LC) rates at 1 and 2 were 86% and 78%, respectively. Female gender, age < 70 years, and tumor size < 5 cm were significantly associated with better OS. The group of patients with tumors close to the trachea but further away from the PBT also correlated with better OS. The acute G2 dysphagia, cough, and dyspnea were 11%, 5%, and 3%, respectively. Acute G3 dyspnea was experienced by one patient. Late G3 toxicity was reported in 4% of patients. Conclusion: risk-adaptive SBRT for ultra-central tumors is safe and effective, even if it remains a high-risk clinical scenario.

## 1. Introduction

Stereotactic body radiotherapy (SBRT) has been the standard option for patients with early stage, medically or surgically inoperable Non-Small Cell Lung Cancer (NSCLC) [1,2]. SBRT has also been adopted for the treatment of inoperable lung metastases [3] in the setting of controlled oligometastatic disease. Overall, SBRT provided excellent outcomes in terms of survival and local control (LC) rates [4], reporting a good tolerability profile in case of peripherally located lung tumors [5]. Notwithstanding the fact that risk-adapted fractionation regimens are widely documented [6,7,8,9], there is still no consensus on optimal treatment schedules and constraints for central tumors. In this regard, the RTOG 0813 trial clarified the definition of central tumors as tumors touching or overlapping a 2 cm zone called the “no-fly zone”, around the proximal bronchial tree (PBT: carina, right and left main bronchi, intermediate bronchus and lobe bronchi) and/or adjacent to the mediastinal pleura [7]. Patients treated with SBRT for central tumors reported a high local control rate with acceptable toxicity rates, as long as fractionation schedules were adequately selected. However, excellent survival outcomes of SBRT can be influenced by significant toxicities related to the high risk setting of central tumors [10]. It is well known that the prescription of 60–66 Gy in 3 fractions for SBRT of central lung tumors leads to higher toxicity rates, if compared to peripheral tumors [11]. Indeed, the prospective experience of Timmerman et al. confirmed 2-year survival rates without grade 3–4 toxicity of 54% and 83% in central and peripheral tumors, respectively. On the other hand, the 5-fractions regimen with a maximum of 12 Gy/fraction, had a grade 3–4 toxicity of 7% and a 2-year local control rate of 88% [7]. Although there was no correlation between radiation pneumonitis and tumor location [12,13], the main complications following the SBRT of central lung tumors [6,7,8,9] were bronchial toxicities. Bronchial stenosis, occlusion, and atelectasis were associated with both the maximum dose (Dmax) and the bronchial volume, receiving more than 65 Gy EQD2 considering an a/b ratio of 3 [14]. As expected, high grade toxicity was more frequently observed in tumors near the main bronchi [15]. In 2015, Chauduri et al. [16] defined for the first time the concept of ultra-central location as a tumor directly touching or overlapping the central airway, such as the trachea and PBT. However, different definitions (Table 1) have been adopted in the published retrospective cohorts of ultra-central tumors, highlighting the lack of consensus among the authors. The present experience aims to identify prognostic factors associated with outcomes for ultra-central tumors. We also evaluated the incidence of acute and long-term toxicity in patients treated with SBRT for ultra-central lung tumors.

## 2. Materials and Methods

### 2.1. Patient Selection

We retrospectively evaluated patients from Erasmus Medical Center (EMC) who were treated between 2006 and 2020 with risk-adaptive SBRT for primary or metastatic ultra-central lung tumors. We included patients with primary tumors diagnosed with T1-4N0 stage of disease and oligometastatic disease, defined as metastases limited to 2 organs and in total, 5 metastases at the time of treatment. According to the multidisciplinary tumor board, patients were considered not suitable for surgery due to severe comorbidity or unresectable disease. Ultra-central tumors were defined as tumors whose planning target volume (PTV) touches or overlaps the trachea, mainstem-, intermediate-, upper-, middle- or lower- lobe bronchus or the esophagus. Oligometastatic patients previously treated with surgery, SBRT, or chemotherapy who developed new oligometastases were eligible. Exclusion criteria were: (1) oligo-progression, (2) M1 stage for patients with primary lung cancer, and (3) prior thoracic field of radiotherapy overlapping the PTV of the current treatment. We received the approval of the Institutional Review Board (IRB) of the EMC [MEC-2016-729].

### 2.2. Treatment Schedules

The procedures and dose fractionation schedules of SBRT used at EMC have been previously reported in the literature [25,26,27,28]. Patients were treated with a Cyberknife^Ⓡ^ Robotic Radiosurgery System (Accuray Inc., Sunnyvale, CA, USA). Dose prescriptions and fractionation schedules ranged from 45 to 60 Gy and from 5 to 7 fractions.

Dose calculations were performed using the Monte-Carlobased algorithm. Ray Tracing dose distributions were recalculated with the Monte-Carlo algorithm [29]. PTV was defined as gross tumor volume (GTV) plus 5 mm. Dose to PTV was prescribed to the 60–87% isodose line covering at least 95% of the PTV. We prioritized the Organ at Risk (OAR) dose constraints over PTV coverage. The patients underwent fiducial-tracking SBRT. If the placement of markers was not possible due to comorbidity or tumor location, we defined an internal-target-volume (ITV).

### 2.3. Follow-Up

Computerized tomography (CT) scan and clinical evaluation were administered 3 weeks after SBRT and 3, 6, 12, 18, 24 months, and then yearly after radiotherapy. We registered treatment related adverse events of the acute and late period defined as within or beyond 4 months, respectively. The severity score used was NCI Common Terminology Criteria for Adverse Events, version 5.0. To determine the toxicity rate, we collected the highest score of acute and late toxicity of each patient.

### 2.4. Endpoints

Overall survival (OS) was defined as the time from start date of SBRT to the date of the last follow up for patients still alive, or the date of death. Progression-free survival (PFS) was defined as time from the date of the first SBRT session to date of local, mediastinal, or distant progression of the disease, whichever occurred first. Living patients without any recurrence on the last follow up were censored. Local control was calculated from the first SBRT to the date of local progression. Patients without local recurrence were censored at the last follow-up date. The toxicity analysis was performed retrospectively. For each patient, we reported the highest score of any kind of acute and late adverse events. 

### 2.5. Evaluation of Prognostic Factors

Variables tested as prognostic factors were age, gender, comorbidity index, Biological Effective Dose (BED), fractionation schedule of SBRT, tumor size, distance from the tumor to the PBT, distance from the tumor to trachea, distance from the tumor to esophagus, and the location of lung tumors (mediastinum, lower lobes, and other locations). Comorbidity was recorded with the Charlson Comorbidity Index (CCI) and the Cumulative Illness Rating Scale (CIRS). 

### 2.6. Statistical Analysis

We assessed survival estimates using the Kaplan–Meier method. Cox regression was used to perform univariable analysis in order to calculate the Hazard Ratio (HR) and 95% confidence interval (CI) for various covariates. The covariates with a *p*-value < 0.20 were used to obtain the multivariable analyses (MVA) by a Cox proportional hazards model. The MVA was calculated using backward linear regression with a threshold of *p* < 0.05. All statistical analyses were performed using IBM SPSS statistics version 22.0 software package (SPSS Inc., Chicago, IL, USA).

## 3. Results

### 3.1. Patient, Treatment and Tumor Characteristics

One hundred twenty-two patients were treated for a total number of 126 ultra-central lung tumors between July 2006 and July 2020. Sixty-eight and 54 patients had a primary tumor or oligometastatic disease to the lung, respectively. Histopathological confirmation of primary tumors and metastases was not mandatory. Fifty-two out of 68 primary tumors were histologically confirmed.

In the majority of patients who underwent SBRT for lung metastasis, the primary tumor locations were breast, lung, and colorectal cancer in 7%, 13%, and 61%, respectively. Only one patient presented with brain metastasis at the time of SBRT for ultra-central tumor, and the main sites of other metastasis were liver and lung.

The median age in the entire cohort was 72 years (range: 34–91 years). Seventy-one patients were male and 51 female. The tumor was located ≤ 5 mm from PBT, the trachea, and the esophagus, in 92, 22, and 41, patients, respectively. The median tumor size was 37.5 mm (range: 8–105 mm). No statistically significant difference (*p* = 0.08) between PTV (mean 53.3 cc; range 5.1–289.0) of patients receiving fiducial-tracking SBRT and PTV of patients treated without markers (mean 64.6 cc; range 10.9–231.7) was reported. Of 126 tumors, 47 (37%) were treated with the SBRT schedule of 7 fractions of 7.5 Gy. The median BED and prescription isodose were 92 Gy (range: 83–132 Gy) and 77% (range: 60–87%), respectively. Patient, treatment, and tumor characteristics are summarized in Table 2.

### 3.2. Survival and Local Control

Median follow-up time was 23 months (range: 0.3–159.4 months). At the time of the analysis, the estimated median OS was 29.3 months (95% CI 22.4–36.21). The 1-,2-, and 5- year OS rates were 75%, 58%, and 23%, respectively. In the entire cohort, 90 patients had died. Estimated median PFS and mean LC were 16 months (95% CI 8.9–23.1) and 62.37 months (95%CI 53.7–71.0), respectively. PFS rates at 1-, 2-, and 5-years rates were 63%, 41%, and 15%, respectively. The 1-, 2-, and 5-year LC rates were 86%, 78%, and 61%, respectively.

### 3.3. Prognostic Factors

Female gender (HR, 0.61 95% CI, 0.4–0.9, *p* = 0.027) was significantly associated with better OS (Table 3). Age ≥ 70 years (HR 1.91, 95% CI 1.2–3.0, *p* = 0.005) and tumor size ≥ 5 cm (HR 1.64, 95% CI 1.1–2.5, *p* = 0.028) correlated with worse OS (Table 3), at univariate analysis. The actuarial OS with these prognostic factors is shown in Figure 1a-c. The influence of a tumor-trachea distance ≤ 5 mm and mediastinal location on OS was also observed at univariate analysis (HR 1.79 95% CI 0.95–3.4, *p* = 0.072 and HR 0.63 95% CI 0.36–1.08, *p* = 0.094, respectively) (Figure 1d,e). The patients with tumors close to the trachea (≤5 mm) but further away from the PBT (>5 mm) were associated with better OS (HR 0.5 95%CI 0.2–1.1, *p* = 0.10) compared to the rest of the cohort (Figure 1f) at univariate analysis. On subset analysis, patients with a tumor size < 5 cm and aged < 70 years experienced significantly better OS than patients with tumor size ≥ 5 cm and aged ≥ 70 years (Figure 2). Univariate analysis showed that no factors were significantly associated with a better LC.

In the multivariate analysis, female gender (HR 0.6 95% 0.4–1.0, *p* = 0.05) resulted as an independent factor significantly associated with a better OS, while age ≥ 70 years (HR 1.9 95% CI 1.2–3.0, *p* = 0.006) and tumor-trachea > 5 mm (HR 2.2 95%CI 1.2–4.2, *p* = 0.015) independently correlated with a worse OS (Appendix A).

The median PFS was 26.2 months (95%CI 0.5–52.0) and 12.4 months (95% CI 7.3–17.5) for the group of primary tumors and oligometastases, respectively. The patients treated for primary tumors showed a significantly better PFS at Kaplan–Meier analysis (*p* < 0.001). Kaplan–Meier curves showed a positive trend for tumor size < 5 cm and PFS (0.068). Moreover, in the subset of patients with primary lung cancer, tumor size < 5 cm showed a significantly better PFS than larger tumors (*p* = 0.004) (Appendix A).

### 3.4. Acute and Late Toxicity

Overall, radiation treatment was very well tolerated and there were no grade (G) 4 or 5 events. Acute G 1-2 toxicities were experienced by 70% of patients. The acute G2 dysphagia, cough and dyspnea rates were 11%, 5%, and 3%, respectively. Acute Grade 3 dyspnea was experienced by one patient. The majority of patients did not develop late toxicity after SBRT treatment. Late G2 events were reported in 10% of patients treated for ultra-central tumors, respectively. Four cases of late G3 dyspnea were reported in the present cohort (Table 4). No G > 2 acute and late esophageal toxicities were observed. At *t*-test analysis, a significant correlation between tumor-esophagus distance and incidence of G 1 or 2 dysphagia (mean 9.2 mm; range 0–40 mm) was reported (*p* < 0.00001), in comparison with the group of patients that did not develop dysphagia (mean 22.65 mm; range 0–65 mm). Any sort of acute or late cardiac events (e.g., pericarditis, myocarditis, or heart failure) were reported.

## 4. Discussion

The present study reports outcomes of a large patient cohort treated with risk-adaptive SBRT for ultra-central lung tumors. The literature still reveals a significant lack of data on SBRT of ultra-central tumors and a need for consensus concerning the definition of ultra-central location, treatment schedules, and constraints. In 2019, Chen et al. conducted a systematic review on the safety and efficacy of SBRT for ultra-central lung tumors, including 10 studies with a total sample size of 250 patients [30]. This overview suggested that SBRT could be a feasible and safe treatment in the high-risk clinical scenario of ultra-central tumors, achieving excellent oncological outcomes with a good tolerability profile [30]. In the last decade, authors reported findings of ultra-central tumors as a separate clinical entity referring to risk of fatal treatment-related toxicity [31]. Indeed, an OS nomogram, developed for medically inoperable centrally located early-stage NSCLC, defined the ultra-central location as a factor associated with worse OS [32]. Therefore, a number of studies investigated a limited sample of patients undergoing SBRT for ultra-central tumors, in comparison with cohorts of patients with central tumors [16,24,33]. SBRT was associated with high rates of LC and no differences between groups of central and ultra-central tumors were described [16,24,33]. 

To the best of our knowledge, this is the retrospective cohort, evaluating exclusively ultra-central tumors, carried out with the largest number of patients. In line with the previously published literature, we showed good 1-year and 5-year LC rates of 86% and 61%, respectively. Median OS was 29 months and 1-year, and 2-year OS rates were 75% and 58%, respectively. Similar results were reported by Breen et al. [21] and Lodeweges et al. [19], with 1-year and 2-year OS rates of 78% and 57%, and 77% and 52%, respectively. One-year LC was 97% in the study reported by Guillaume et al. [17], and 98% in the work by Lodeweges et al. [19], slightly higher than our findings. Across the studies investigating SBRT for ultra-central tumors, there was a significant heterogeneity in terms of fractionation schedules. Indeed, most of them reported a prescribed dose ranging from 50 Gy to 60 Gy, delivered with 5 to 15 fractions of 5 to 12 Gy [16,34,35]. In the present study, the entire cohort was treated with CyberknifeⓇ Robotic Radiosurgery System (Accuray Inc, Sunnyvale, CA), and primarily by fiducial-tracking SBRT technique (75%). The majority of patients received a risk-adapted SBRT based on 7 fractions of 7.5 Gy with a median BED of 92 Gy, within a range from 83 to 132 Gy. A commonly adopted SBRT schedule for ultra-central tumors was 60 Gy delivered in 8 fractions of 7.5 Gy and BED ranged from 75 to 132 [23,36].

According to the previous literature [16,17,24], no grade 4 or 5 with a very low rate of acute (1%), and late (4%) G3 events, were reported [19]. Acute and late G3 dyspnea were experienced by 1 and 4 patients, respectively. Concerning cardiac toxicity, we did not register cases of pericarditis, myocarditis, or other radiation-related heart injuries. However, a longer median follow up is needed to assess late cardiac toxicities related to SBRT for ultra-central lung tumor.

On the other hand, few studies showed a SBRT-related toxicity higher than the one presented in the current analysis. Grade ≥ 3 and 5 rates were 22% and 11% in the cohort of Wang et al., although 10% of patients underwent antiangiogenic therapies within 3 months of SBRT [37,38]. Similarly, Tekatli et al. reported a high rate of ≥ G 3 (38%) and G 5 (21%) toxicities [9]. Of the 65 patients included in the recent Hilus trial [6], 7 experienced acute G3 dyspnea and 8 bronchopulmonary hemorrhage. Patients in the aforementioned study received 56 Gy in 8 fractions and most toxicity was found in the group of tumors ≤ 1 cm from main bronchi and trachea. 

The different rates of treatment-related toxicity can be explained by the decision of some authors to strictly prioritize the respect of OARs dose constraints rather than the PTV coverage [16,17,24] in order to avoid high risk of acute toxicity such as dysphagia, esophagitis, and airway bleeding. Moreover, adequately identifying critical normal structures is a mandatory step of the treatment planning process which was missing in the HILUS trial [39]. The consequent wide variability of the PBT contouring could contribute to significant hotspots in crucial substructures, such as lobar or main bronchi [39]. The novelty of our study is the purpose to define prognostic factors for survival in a homogenous population of exclusively ultra-central lung tumors, which has been rarely investigated over the past years [21]. 

To our knowledge, most of the published studies investigating only ultra-central lung tumors analyzed the high-risk indicators for treatment-related mortality or local control, excluding clinical factors significantly associated with better OS or PFS, except for the study of Breen et al. [21]. Interestingly, unlike previous literature [17,19,30], this study identified female gender, age < 70 years, and tumor-trachea distance ≤ 5 mm as independent prognostic factors associated with better OS. Moreover, patients with a tumor size < 5 cm were significantly associated with better OS as compared to patients with a tumor size ≥ 5 cm. Likewise, higher age, disease stage, and bigger PTV were associated with worse OS in one of the largest cohorts of central and ultra-central NSCLC patients (220) treated with SBRT [32]. Furthermore, the distance from the tumor to ultra-central OARs (PBT, trachea, and esophagus) was evaluated in few studies without significant correlation with survival [6]. In the present cohort, mediastinal location was described in 25 patients. Patients with a tumor-PBT and tumor-trachea distance ≤ 5 mm were 73% and 17%, respectively. Only 7 patients received SBRT for tumors with a distance ≤ 5 mm from both PBT and trachea. Of note, patients with a tumor-trachea ≤ 5 mm had a better OS. This fact could be related to a higher distance from the PBT and hilus for the majority of patients with a tumor-trachea distance ≤ 5 mm, and a consequent lower risk of metastasizing. Indeed, Breen et al. analyzed one of the largest samples of patients treated with SBRT for ultra-central tumors (110), reporting that a shorter distance from GTV to PBT was a factor associated with death [21]. In our cohort of patients, tumor-PBT distance ≤ 5 mm was associated with worse OS, albeit the data were not statistically significant (*p* 0.13). Moreover, the subgroup of patients with tumors close to the trachea (≤ 5 mm) and further away from the PBT (> 5 mm) (14) reported a better OS (*p* = 0.098). Accordingly, the subset of patients with a tumor size < 5 cm and a tumor-PBT distance ≤ 5 mm experienced significantly better OS than patients with a tumor size ≥ 5 cm. The subset of patients with tumor sizes < 5 cm confirmed that age < 70 years was a prognostic factor significantly associated with a better OS. 

The limit of our study is the retrospective nature of the analysis and the adoption of different fractionation schedules due to the change of clinical practice over a wide period of 14 years. 

## 5. Conclusions

Risk-adaptive SBRT for ultra-central tumors is safe and effective, even if it remains a high-risk clinical scenario. Patients with larger tumor sizes, who were elderly, and male, treated for ultra-central lung tumors, reported a worse OS. Moreover, tumor locations further away from the hilus of the lung seems to be a prognostic factor associated with a better outcome. Indeed, patients should be carefully selected on the basis of various ultra-central sites and patient characteristics. Practice guidelines for SBRT of ultra-central tumors are currently missing and the present findings could be used to support clinical decision making. However, further dosimetric analyses are still warranted to establish the optimal SBRT regimen that could be delivered without high risks of severe toxicity. 

## Figures and Tables

**Figure 1 cancers-14-05908-f001:**
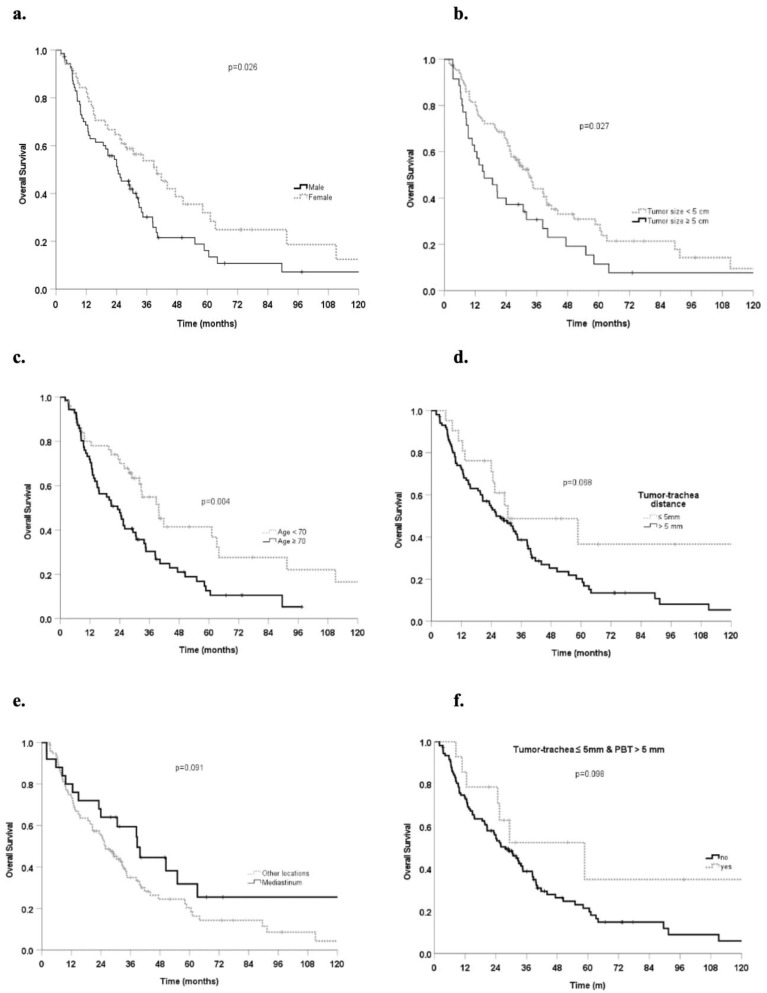
(**a**) Overall survival for male and female patients, (**b**) patients with tumor sizes < or ≥5 cm, (**c**) aged< or ≥70 years, (**d**) with tumor-trachea distance≤ or >5 mm (**e**) treated for mediastinal tumors vs. the other locations and (**f**) patients with distance from tumor to the trachea ≤ 5 mm and to the PBT > 5 mm.

**Figure 2 cancers-14-05908-f002:**
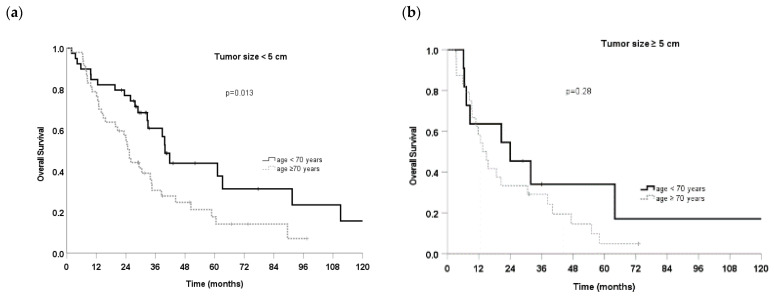
Overall survival for patient with tumor sizes (**a**) < or ≥5 cm (**b**) and age< vs. ≥70 years.

**Table 1 cancers-14-05908-t001:** Ultra-central definitions.

Study, Year	N of Patients	Definition of Ultra-Central
Guillaume, 2021 [17]	74	PTV overlapping trachea, right and left main bronchi, intermediate bronchus, lobar bronchi, esophagus, heart.
Lindberg, 2021 [6]	65	1-cm zone around the PBT.
Farrugia, 2021 [18]	43	GTV abutting the PBT, trachea, mediastinum, aorta, or spinal cord.
Lodeweges, 2021 [19]	72	PTV abutting or overlapping the main bronchi, trachea and/or esophagus
Mihai, 2021 [20]	57	GTV abutting or involving trachea, main or lobar bronchi.
Breen, 2021 [21]	110	GTV directly touching the PBT or trachea. PTV overlapping the trachea or mainstem bronchi.GTV within 1 cm of the PBT.
Regnery, 2021 [22]	51	Overlap of the PTV with the PBT
Loi, 2020 [23]	109	PTV overlapping with central bronchial tree, esophagus, pulmonary vein, or pulmonary artery.
Zhao, 2020 [24]	98	PTV overlapping with PBT, esophagus, pulmonary vein or pulmonary artery.

Abbreviation: GTV, gross tumor volume; PTV, planning tumor volume; PBT, proximal bronchial tree.

**Table 2 cancers-14-05908-t002:** Patient, treatment and tumor characteristics.

Characteristics			n	Percentage
Age (median, range)	72 years	(34–91)		
<70 years			51	42%
≥70 years			71	58%
Gender				
Male			71	58%
Female			51	42%
Comorbidity index				
CCI (median, range)	1	(0–8)		
Score 0			27	22%
Score 1–2			56	46%
Score ≥ 3			39	32%
CIRS (median, range)	3	(0–16)		
Score 0–4			90	74%
Score ≥ 5			32	26%
Tumor-PBT distance				
≤5 mm			92	73%
>5 mm			34	27%
Tumor-esophagus distance				
≤5 mm			41	33%
>5 mm			85	67%
Tumor-trachea distance				
≤5 mm			22	17%
>5 mm			104	83%
Tumor size (median, range)	37.5 mm	(8–105)		
≤5 cm			90	71%
>5 cm			36	29%
Tumor location				0%
Mediastinum			28	22%
Lower lobe			46	37%
Other lobes			52	41%
Dose fractionation schemes				
7 fractions of 7 Gy **			25	20%
7 fractions of 7.5 Gy *			47	37%
7 fractions of 8 Gy **			8	6%
6 fractions of 8 Gy **			19	15%
5 fractions of 9 Gy **			4	3%
5 fractions of 10 Gy or 12 Gy **			6	5%
5 fractions of 11 Gy *			17	13%
Markers	3	(0–16)		
yes			95	75%
no			31	25%
BED (median, range)	92 Gy	(83–132 Gy)		
<100 Gy			91	72%
≥100 Gy			35	28%
Prescription isodose (median, range)	77%	(60–87%)		
SBRT target				
Primary lung tumors			68	54%
Lung metastasis			58	46%

Abbreviations: CCI Charlson Comorbidity Index; CIRS, Cumulative Illness Rating Scale; PBT, proximal bronchial tree; BED, Biological Effective Dose; SBRT, Stereotactic body radiotherapy. * Monte-Carlo calculation. ** Ray Tracing calculation.

**Table 3 cancers-14-05908-t003:** Prognostic factors at univariate analysis.

Covariates	Median OS Months (95%CI)	HR (95%CI)	*p* Value
SBRT target			
Lung metastasis (54)	34.4 (26.1–42.8)	0.69 (0.45–1.1)	0.089
Primary lung tumors (68)	20.6 (9.6–31.6)		
Gender			
Female (51)	39.8 (29.3–53.4)	0.61 (0.4–0.9)	0.027
Male (71)	24.4 (16.4–32.5)		
Age			
≥70 years (71)	23.2 (14.0–32.4)	1.91 (1.2–3.0)	0.005
<70 years (51)	39.7 (31.1–48.4)		
Tumor size			
≥5 cm (36)	15.5 (6.5–24.6)	1.64 (1.1–2.5)	0.028
<5 cm (86)	33.0 (27.4–38.5)		
Tumor-PBT distance			
>5 mm (32)	33.8 (22.4–45.3)	0.68 (0.4–1.1)	0.126
≤5 mm (90)	25.9 (18.9–32.9)		
Tumor-esophagus distance			
>5 mm (84)	25.9 (20.3–31.4)	1.36 (0.86–2.15)	0.182
≤5 mm (38)	33.9 (22.8–44.9)		
Tumor-trachea distance			
>5 mm (101)	26.0 (18.3–33.7)	1.79 (0.95–3.4)	0.072
≤5 mm (21)	30.7 (0.0–61.7)		
Tumor-trachea ≤ 5 mm &PBT> 5 mm distance			
Yes (14)	58.8 (18.4–99.1)	0.53 (0.2–1.14)	0.104
No (108)	27.7 (20.3–35.1)		
CCI			
Score ≥ 3 (39)	25.9 (18.9–32.8)	1.1 (0.7–1.6)	0.84
Score < 3 (83)	30.6 (22.8–38.5)		
CIRS			
Score ≥ 5 (32)	14.9 (5.1–24.8)	1.2 (0.7–1.9)	0.45
Score < 5 (90)	32.7 (25.1–40.2)		
Tumor location			
Other lobes (76)	25.9 (19.4–32.3)	1.0 (0.7–1.6)	0.93
Lower lobe (46)	31.9 (25.9–37.9)		
Mediastinum	38.8 (26.1–51.5)	0.63 (0.36–1.08)	0.094
Other locations	25.9 (20.1–31.6)		
BED			
≥100 (31)	25.4 (22.9–27.9)	1.1 (0.7–1.7)	0.69
<100 (91)	30.5 (22.7–38.2)		
Markers			
no (29)	30.6 (22.3–39.0)	0.7 (0.4–1.2)	0.21
yes (93)	26 (12.0–40.0)		

Abbreviations: CCI Charlson Comorbidity Index; CIRS, Cumulative Illness Rating Scale; PBT, proximal bronchial tree; bed, Biological Effective Dose; SBRT, Stereotactic body radiotherapy.

**Table 4 cancers-14-05908-t004:** SBRT related toxicity.

Acute Toxicity	Pain	Dyspnea	Cough	Fatigue	Dysphagia	Highest Toxicity
G0	113 (93%)	101 (83%)	85 (70%)	80 (66%)	83 (68%)	36 (30%)
G1	8 (7%)	16 (13%)	30 (25%)	35 (29%)	27 (22%)	59 (48%)
G2	1 (1%)	4 (3%)	6 (5%)	6 (5%)	13 (11%)	26 (21%)
G3	0 (0%)	1 (1%)	0 (0%)	1 (1%)	0 (0%)	1 (1%)
Late toxicity	Pain	Dyspnea	Cough	Fatigue	Dysphagia	Highest toxicity
G0	115 (94%)	99 (81%)	100 (82%)	103 (84%)	119 (98%)	79 (65%)
G1	2 (2%)	12 (10%)	17 (14%)	13 (11%)	2 (2%)	26 (21%)
G2	4 (3%)	7 (6%)	5 (4%)	6 (5%)	1 (1%)	12 (10%)
G3	1 (1%)	4 (3%)	0 (0%)	0 (0%)	0 (0%)	5 (4%)

## Data Availability

The data that support the findings of this study are available from the corresponding author.

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
