# Peer review of "Survival and Prognostic Factors of Ultra-Central Tumors Treated with Stereotactic Body Radiotherapy"

_cancers, 2022, doi:10.3390/cancers14235908_

Round 1
Reviewer 1 Report
General Comments
The manuscript is well written. I have a question regarding esophagitis. The authors did not report any such toxicity. May I ask why?
The toxicities reported seems to cover airway related ones only. Can you expand the study to cardiac besides airway related toxicities?
Specific Comments
Table 2: Should you report the distance to cardiac structures? Tumor to esophagus distance was not used to report esophagitis.
Reviewer 2 Report
In this paper, the authors have analyzed the treatment outcomes and toxicities of risk-adaptive SBRT for ultra-central tumors. It has been demonstrated that patients with larger tumor size, elderly and male patients treated for ultra-central lung tumors report a worse significant overall survival and the group of patients with tumor close to trachea but further away from proximal bronchial tree were correlated with better OS.
However, their data and explanation raise some questions and critics.
1. In materials and methods, the authors defined oligometastatic disease as metastases limited to 2 organs. Were patients with brain metastases included in analysis? If so, how many? This is important information because the prognosis can completely different when there is brain metastasis.
2. If previous treatment with SBRT were allowed, were there patients who underwent re-irradiation included in analysis? Please specify in detail. If so, how was it taken into consideration when prescribing the dose?
3. The authors have mentioned that in 25% of patients who did not undergo fiducial insertion, ITV was defined. Was there any difference in median PTV volume depending on whether or not fiducial was performed? If there is so, is it correlated with treatment outcomes or toxicities? Please add the results to the manuscript.
4. Were how many patients treated without histopathologic confirmation? This can be an important information since the treatment effect may have been overestimated.
5. Please indicate what the primary cancers of 54 patients who underwent SBRT for lung metastasis were and what their rates were. The survival outcomes can vary greatly depending on what the primary cancer is.
6. In results, equal and inequality signs for p values are missing. Please insert the signs correctly.
7. In figure 1, the figures seem to be out of order. Please arrange them in order correctly.
8. The authors have identified the prognostic factors associated with better survival and also demonstrated that the group of patients with tumor close to trachea but further away from proximal bronchial tree was correlated with better OS. How do you think this result will affect your actual practice in terms of patient selection and dose prescription in the future?
Round 2
Reviewer 1 Report
Thanks for the revision